nanotechnology

antibacterial activity, *Staphylococcus aureus*, silver nanoparticles, amphiphilic star block copolymers

**Author for correspondence:**
Muhammad Imran Malik
e-mail: mimran.malik@iccs.edu

This article has been edited by the Royal Society of Chemistry, including the commissioning, peer review process and editorial aspects up to the point of acceptance.

# Enhancement in the antibacterial activity of cephalexin by its delivery through star-shaped poly(ε-caprolactone)-block-poly(ethylene oxide) coated silver nanoparticles

Sana Rahim, Samina Perveen, Shakil Ahmed,
Muhammad Raza Shah and Muhammad Imran Malik

H.E.J. Research Institute of Chemistry, International Centre for Chemical and Biological Sciences (ICCBS), University of Karachi, Karachi 75270, Pakistan

MIM, 0000-0001-6942-0407

The antibacterial activity of silver nanoparticles (AgNPs) stabilized with a four-armed star-shaped poly(ε-caprolactone)-block-poly(ethylene oxide) copolymer [St-P(CL-b-EO)] and its application as a drug delivery vehicle for cephalexin (Cp) was evaluated against pathogenic *Staphylococcus aureus*. The prepared AgNPs were characterized by ultraviolet-visible spectroscopy, Fourier transform infrared spectroscopy, zeta sizer and atomic force microscopy (AFM). The antibacterial efficiency of Cp is enhanced several-fold by its delivery through complexation with St-P(CL-b-EO)-AgNPs, monitored by microplate assay and biofilm destruction studies. Finally, the visual destruction of bacterial cells and its biofilms by employing Cp and its conjugates at their minimum inhibitory concentration ($MIC_{50}$) and minimum biofilm inhibitory concentration ($MBIC_{50}$), respectively, is observed by topographic imaging by AFM. Enhanced antibacterial activity of St-P(CL-b-EO)-AgNPs loaded Cp is attributed to penetrative nature of the drug cargo St-P(CL-b-EO)-AgNPs towards the bacterial cell wall.

## 1. Introduction

Infectious diseases are one of the major reasons of morbidity and mortality across the world [1–3]. According to an estimate, 20–30

per cent of the human population is a longstanding carrier of *Staphylococcus aureus*, a round-shaped gram-positive bacterium that belongs to the phylum Firmicutes. *Staphylococcus aureus* is a natural inhabitant of the normal skin flora in the nostrils and the lower genital part of a women's reproductive system. *Staphylococcus aureus* causes many illnesses, including skin infections such as carbuncles, abscesses, pimples, cellulitis and scalded skin syndrome such as osteomyelitis, sepsis, endocarditis, pneumonia, bacteremia and toxic shock syndrome [4–6]. These pathogenic strains mostly spread infections via the production of potent protein toxins and the cell surface protein expression that inactivates antibodies through binding. In clinical medicines, the development of antibiotic-resistant strains of *S. aureus*, i.e. methicillin-resistant *S. aureus*, is a global problem. Despite extensive research in the field, no immunization is yet approved against *S. aureus* [3,7–9].

Moreover, the infections caused by *S. aureus* are generally treated by cephalexin (Cp) [10,11]. Cp is a first-generation cephalosporin that fits in the class β-lactam antibiotics and exhibits enormous antibacterial activity against both Gram-negative and Gram-positive bacteria [12,13]. However, bacterial communities, especially *S. aureus*, have shown resistance to such traditional antibiotics [11]. Hence, continuous research on the production of active antibacterial agents is extensively required because of the advent of multi-drug-resistant bacterial strains, along with efforts to enhance the efficacy of the existing drugs.

Initial focus of research on nano–biointeractions include studies comprised both the biological compositions as well as material properties, such as analysis of biomolecular signalling, chemical functionality, clearance, transport kinetics, gene expression variations [14], surface charge and toxicity [15]. In this context, the term 'polymer therapeutics' encompasses polymer-based drugs, polymer protein conjugates, block copolymer micelles, aptamers and multicomponent nonviral vectors with covalent linkages [16]. Polymers allow flexibility and versatility in therapeutic applications owing to the possible modifications based on their functionalization, different synthesis methods, number of variable distributions and diversity [17–19]. However, the application of large-sized materials such as polymers in drug delivery has to encounter numerous challenges, for instance poor solubility, poor bioavailability, *in vivo* instability, poor absorption in the body, issues with target-specific delivery and tonic effectiveness, beside its possible unfavourable effects on the drug efficacy. In this regard, novel drug delivery systems such as polymer-stabilized metal nanoparticles could provide a synergistic combination of properties for targeted drug delivery [20,21]. Furthermore, nanoparticle-based antimicrobial formulation has appeared to act as an effective antibactericidal material. These nanoparticles are sub-colloidal structures (size: 10–100 nm) that serve as efficient drug carriers for enhancement in the drug potency [22,23]. Several authors have presented research articles and reviews signifying the importance, synthesis, characterization and myriad of applications of metal nanoparticles in the modern technological world [24–29].

Silver nanoparticles (AgNPs) with well-known surface chemistry, chemical stability and controlled geometry have been employed for biological applications [30,31]. The enhancement in the antibacterial potential of ampicillin was shown by its combination with gold and silver NPs versus multi-drug-resistant strains including *S. aureus, Enterobacter aerogenes and Pseudomonas aeruginosa* [32]. Naked AgNPs are prone to aggregation, hence surface stabilization of nanoparticles is imperative. The potential of a star-shaped poly(ε-caprolactone)-block-poly(ethylene oxide) copolymer [St-P(CL-b-EO)] as a stabilizing agent for AgNPs is demonstrated in a recent study by our research group [33]. St-P(CL-b-EO), a non-toxic and non-ionic polymer, exhibits excellent properties of biodegradability, biocompatibility and easy excretion from the body. Furthermore, ether and ester groups of St-P(CL-b-EO) contain an oxygen atom that has an affinity to chelate with metal ion which ultimately induces stabilization and prevents aggregation of metal NPs [34–37].

In this study, we report the preparation of AgNPs stabilized with four-armed St-P(CL-b-EO) and studied its proficiency to boost the antibacterial potential of Cp through the determination of minimum inhibitory concentration (MIC) and minimum biofilm inhibitory concentration (MBIC). The overall purpose of the study is to evaluate the potential of complexation of AgNPs with Cp in context of enhancement in its therapeutic efficacy.

# 2. Experimental

## 2.1. Materials and instruments

ε-Caprolactone (ε-CL) and sodium borohydride (NaBH$_4$) were purchased from TCI, Japan; methoxy poly(ethylene oxide) (MeO-PEO), maleic anhydride, stannous octoate (Sn(Oct)$_2$), silver nitrate (AgNO$_3$) and pentaerythritol were purchased from Aldrich, Germany; 4-dimethylaminopyridine was

purchased from Across Organics, USA; and trimethylamine by Daijung, Korea were used as received. High performance liquid chromatography grade solvents, chloroform and acetone from RCI Labscan limited, Thailand were used as the reaction medium and for cleaning of apparatus.

Glassware was cleaned with aqua regia to avoid metal contamination, dried in an oven and rinsed with deionized water followed by acetone before use.

A digital pH meter (model 510, Oakton, Eutech) was employed for monitoring the pH of St-P(CL-b-EO)-AgNPs solution. Ultraviolet (UV)-visible spectroscopy was performed on a Shimadzu double beam spectrophotometer, UV-1800 series, operated in the wavelength range of 800–190 nm. A quartz cuvette having path length of 1 cm was used for screening of samples. Bruker Vector 22 Fourier transform infrared spectroscopy (FTIR) spectrometer was used to record FTIR spectra using potassium bromide pellets in the infrared range (400–4000 cm$^{-1}$). Ten scans were used in order to get 0.1 cm$^{-1}$ spectral resolution.

Surface morphology of St-P(CL-b-EO)-AgNps was studied by an Agilent 5500 atomic force microscope, (USA). A drop of St-P(CL-b-EO)-AgNps was dropped on the silicon wafer and dried for 24 h in an inert atmosphere. A zeta sizer (Nano-ZSP, Malvern Instruments) was used to determine particle size distribution and zeta potential of the St-P(CL-b-EO)-AgNps along with their complex with Cp. The study was carried out at a scattering angle of 90° at 25°C. Disposable cuvettes were used to determine zeta size while zeta potential studies were carried out in dip cell cuvettes.

## 2.2. Synthesis of four-armed St-P(CL-b-EO) copolymer

Ring-opening polymerization was employed to synthesize four-armed star-shaped polycaprolactone using pentaerythritol as an initiator and stannous octoate as a catalyst. In parallel, the hydroxyl end-functionality of MeO-PEO is changed to a carboxylic (-COOH) group. Finally, both –OH terminated star-shaped polycaprolactone (PCL) and –COOH terminated MeO-PEO moieties were joined to form a four-armed St-P(CL-b-EO) copolymer. The detailed synthesis protocol and characterization is particularized in our previous publication [38]. The St-P(CL-b-EO) used for this study was a coupled product of star-PCL$_{10}$K and MeO-PEO$_2$K.

## 2.3. Synthesis of St-P(CL-b-EO)-silver nanoparticles

St-P(CL-b-EO)-AgNPs were prepared in a mixture of solvents, composed of acetone and water [33]. The concentration of AgNO$_3$, St-P(CL-b-EO) and NaBH$_4$ were 1.0, 0.1 and 4.0 mM, respectively. One millilitre of St-P(CL-b-EO) solution was added in 50 ml aqueous AgNO$_3$ (an optimized ratio). NaBH$_4$ (0.1 ml) was added after 15 min of constant stirring. The colourless reaction mixture turned yellow after 30 min of continuous stirring and indicates the successful synthesis of AgNPs. Detailed characterization of synthesized St-P(CL-b-EO)-AgNPs is particularized in our previous work [33].

## 2.4. Preparation of St-P(CL-b-EO)-silver nanoparticles/cephalexin complex

A 0.1 mM solution of Cp was prepared in distilled water and mixed with equal volume of St-P(CL-b-EO)-AgNPs (1–50) solution to make a St-P(CL-b-EO)-AgNPs/Cp complex. The complex formation was confirmed by UV-visible spectroscopy, atomic force microscopy (AFM), zeta sizer, zeta potential and FTIR as reported in the earlier publication of this series [33].

## 2.5. Determination of loading efficiency

Loading efficiency of the drug in the above-mentioned compelxation is determined through a UV-visible spectrophotometer (Shimadzu, UV-240, Hitachi U-3200). In brief, the above-mentioned formulation of drug and nanoparticles was centrifuged at 14 000 r.p.m. for 25 min. The pellets at the bottom of the Eppendorf tube were carefully collected, followed by the dissolution of these pellets in acetone. In parallel, a calibration curve for the amount of Cp in acetone is constructed by recording UV-visible spectra at 266 nm of solutions of different concentrations of Cp (0.1 to 0.5 mg ml$^{-1}$) which is then employed for the determination of free drug in acetone. The encapsulation efficiency of micelles was calculated by using the following equation:

$$\%\text{encapsulation efficiency} = \frac{A_{\text{(Total drug)}} - A_{\text{(Free drug)}}}{A_{\text{(Total drug)}}} \times 100,$$

where $A$ is the amount of the drug.

## 2.6. Antibacterial assay

### 2.6.1. Bacterial strains

*Staphylococcus aureus* ATCC 25923, a Gram-positive bacteria, was selected for antibacterial assay. To prepare the stock culture of the bacterial strain, it was kept on tryptic soya agar (Oxford, UK) at 4°C. Initially, the bacterial culture was sub-cultured on a fresh agar plate for 24 h before subjecting it to antibacterial assay. Inocula of *S. aureus* was arranged by vaccinating multiple distinct colonies of bacteria into a liquefied medium of Mueller Hinton broth. The bacterial suspension finally gets homogeneous having a final density of $1 \times 10^6$ cfu ml$^{-1}$ that was confirmed by viable counts of colonies.

### 2.6.2. Microplate assay for growth inhibition efficiency

The bacterial growth inhibition efficiency of the test sample and reference materials, Cp, St-P(CL-b-EO), St-P(CL-b-EO)-AgNPs and St-P(CL-b-EO)-AgNPs/Cp were assessed by 96-well microplate assay. The growth reduction of bacterial cells was quantified through tetrazolium dye reduction methods by a microtiter plate reader (Spectramax) [39]. A concentration of $10^6$ cfu ml$^{-1}$ of freshly collected bacterial cell suspension of *S. aureus* was sown in each well of a 96-well plate. A serial dilution of the above-mentioned materials was conducted in Muller Hinton broth in a range of 10 to 500 μg. In triplicate wells, 200 μl of each concentration was placed and plates were incubated under constant shaking of 150 rpm at 37°C ± 0.5 for 18–24 h. After incubation, 50 μl of 3-(4, 5-dimethylthiazol-2-yl)-2, 5-diphenyltetrazolium bromide (MTT dye) solution (0.2 mg ml$^{-1}$) was added in each well of the 96-well plate, and the plate was again incubated under similar conditions for 30 min. The negative control was taken as dimethyl sulfoxide (DMSO), while positive control was conducted through bacterial suspension. The absorbance was measured at 570 nm by adding DMSO on a spectrophotometer while reference wavelength was 650 nm. The per cent reduction of MTT dye was calculated to indicate the bacterial growth inhibition [40]:

$$\text{growth inhibtion (\%)} = \left[ \frac{\text{optical density of control} - \text{optical density of test}}{\text{optical density of control}} \right] \times 100.$$

### 2.6.3. Biofilm growth inhibition efficiency

Antibiofilm activity of *S. aureus* was evaluated by using the microtiter biofilm plate method against four selected combinations of Cp, St-P(CL-b-EO), St-P(CL-b-EO)-AgNPs and St-P(CL-b-EO)-AgNPs/Cp. The compounds were diluted in 96-well flat-bottom plates (Corning, USA). An inoculum consisting of $1 \times 10^6$ cfu ml$^{-1}$ of bacterial cells were simply inoculated in each well except broth which was taken as a negative control. The plates were incubated at 37°C overnight. To check perfect biofilm formation, the plates were stained with 0.1% (w/v) crystal violet for 20 min after washing thrice with sterile distilled water to remove planktonic cells. The stained plates were rewashed with sterile distilled water and the retained crystal violet-stained biofilms were dissolved in 30% (v/v) glacial acetic acid. The microtiter plate reader (Tecan, USA) was set at 590 nm to take absorbance of plates. Percentage biofilm inhibition was calculated by the following formula:

$$\text{biofilm growth inhibtion (\%)} = \left[ \frac{\text{optical density of control} - \text{optical density of test}}{\text{optical density of control}} \right] \times 100.$$

### 2.6.4. Atomic force microscopy

Fresh culture of *S. aureus* was grown up in tryptic soy agar (Oxoid) for 24 h at 37°C. Newly prepared mica slides by cleaving are arranged by adding 10 μl of poly L-lysine which was dried at ambient temperature under the sterilized air flow chamber. A few drops of diluted culture of bacterial strain with the concentration of $10^6$ cfu ml$^{-1}$ were applied on already prepared poly L-lysine mica slides. Five to 10 μl of tested compound samples were also introduced from the respective wells of microtiter plates for inhibition and biofilm inhibition evaluations at their respective MIC$_{50}$ and MIBC$_{50}$. All the slides were dried at ambient temperature under sterile conditions and imaged by AFM for morphological characteristic evaluation of the bacterial cells.

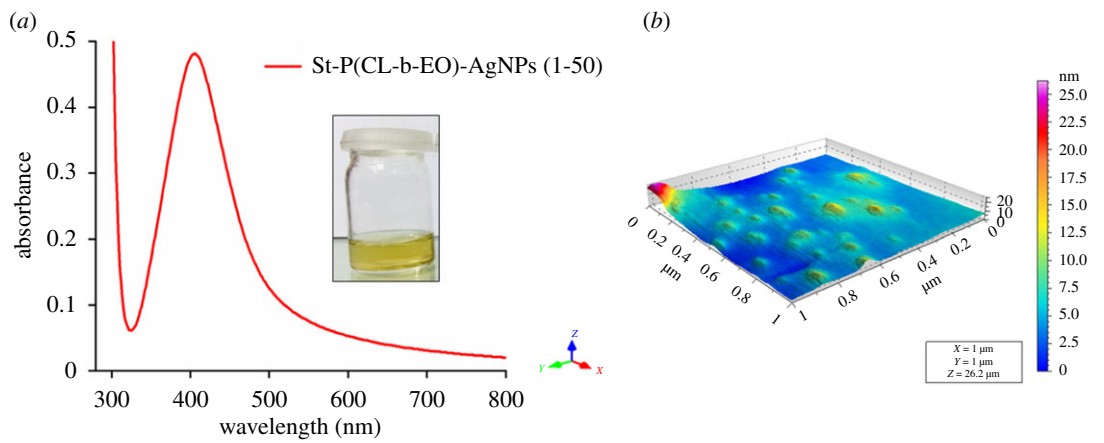

**Figure 1.** Confirmation of formation of St-P(CL-b-EO)-AgNPs: (a) UV-visible spectrum; (b) AFM topographic image.

# 3. Results and discussion

## 3.1. Molecular characterization of St-P(CL-b-EO)

As mentioned in the experimental section, St-P(CL-b-EO) was synthesized by coupling of St-PCL$_{10}$K with MeO-PEO$_2$K. The average number and average weight molar mass of synthesized four-armed St-PCL$_{10}$K as obtained by polystyrene calibrated size exclusion chromatography was 9700 and 13 700 g mol$^{-1}$. MeO-PEO$_2$K used in the coupling reaction was a commercial product. The four-armed St-P(CL-b-EO) were comprehensively characterized by size exclusion chromatography, FTIR, $^1$H-NMR and liquid chromatography at critical conditions of both PCL and PEO. Size exclusion chromatography (SEC) elugrams and $^1$H-NMR spectra of precursors MeO-PEO$_2$K, and St-PCL$_{10}$K, along with the final product St-P(CL$_{10 K}$-b-EO$_2$K) are presented in the electronic supplementary material, figures S1 and S2. The comprehensive characterization of intermediates and final product clearly demonstrate the successful synthesis which is elaborated in great detail in [38].

## 3.2. Synthesis and characterization of St-P(CL-b-EO)-silver nanoparticles

St-P(CL-b-EO) is a four-armed star-shaped block copolymer comprising of PEO as the hydrophilic segment and PCL as the hydrophobic part. St-P(CL-b-EO)-AgNPs were synthesized by NaBH$_4$ reduction method of AgNO$_3$ in the presence of St-P(CL-b-EO). A characteristic surface plasmon resonance (SPR) band at 415 nm was obtained in the UV-visible spectrum by mixing AgNO$_3$ and St-P(CL-b-EO) solutions at an optimized ratio of 50 : 1 (v/v) for 30 min, figure 1a. AFM analysis revealed that St-P(CL-b-EO)-AgNps are polydisperse and have size in the range of 5–20 nm figure 1b). The inset depicts the pale yellowish colour of stabilized St-P(CL-b-EO)-AgNPs.

The stability of St-P(CL-b-EO)-AgNPs against different external parameters, such as temperature, and the presence of electrolytes and pH are evaluated. Temperature treatment of synthesized St-P(CL-b-EO)-AgNPs at 100°C for 20 min resulted in an increase in the intensity of the SPR band which is an indication of enhanced stabilization (figure 2). Moreover, St-P(CL-b-EO)-AgNPs persisted for more than 12 months at ambient temperature.

To check the stability of St-P(CL-b-EO)-AgNPs in the presence of electrolytes, various electrolyte concentrations (0.01 M–5 M NaCl) were added in St-P(CL-b-EO)-AgNPs and their stability is monitored by UV-visible spectroscopy, figure 3. A visible decrease in the sharpness of the absorbance peak is observed by an increase in the electrolyte concentration. Moreover, the typical AgNPs peak just appeared as a shoulder by addition of NaCl beyond 1.0 M. The aggregation of AgNPs by addition of electrolyte is attributed to the presence of a large amount of Cl$^{-1}$ ions in the solution [41,42].

Synthesized St-P(CL-b-EO)-AgNPs were slightly acidic in nature having a pH approximately 5. The changes in the pH in a range of 2–12 resulted in a visible reduction in the intensity of the SPR band of St-P(CL-b-EO)-AgNPs, figure 4. It is found that synthesized St-P(CL-b-EO)-AgNPs have maximum stability at the original pH (approx. 5), as deduced from the intensity of the SPR band. Furthermore, the stability of AgNPs decreased while moving away from original pH, which is more pronounced towards the acidic environment compared to neutral or slightly basic pH.

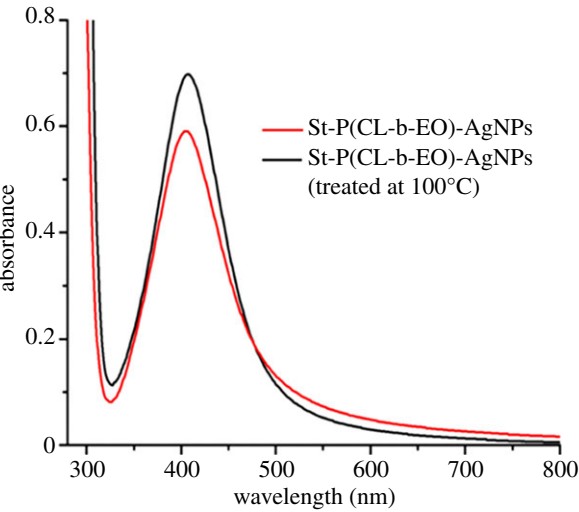

**Figure 2.** UV-visible spectrum of St-P(CL-b-EO)-AgNPs after incubation at 100°C for 20 min.

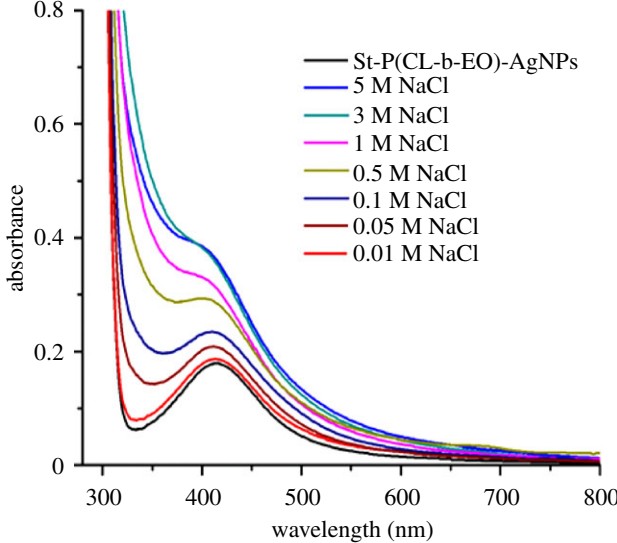

**Figure 3.** Change in the SPR band of St-P(CL-b-EO)-AgNPs as a function of NaCl concentration (0.01 M-5 M).

## 3.3. Complexation of St-P(CL-b-EO)-silver nanoparticles with cephalexin

The addition of Cp in the St-P(CL-b-EO)-AgNPs resulted in the formation of a complex that is more stable compared to parent St-P(CL-b-EO)-AgNPs. The size of St-P(CL-b-EO)-AgNPs/Cp complex decreased (dynamic light scattering and AFM analysis), while stability increased (zeta potential values) compared to St-P(CL-b-EO)-AgNPs [33]. The average size of St-P(CL-b-EO)-AgNPs decreased from $151.5 \pm 8.071$ to $140.5 \pm 9.895$ nm for St-P(CL-b-EO)-AgNPs/Cp. Similarly, zeta potential of values of $-3.33 \pm 4.18$ for St-P(CL-b-EO)-AgNPs compared to $-10.6 \pm 4.27$ mV for St-P(CL-b-EO)-AgNPs/Cp indicates enhanced stability of the latter. Figure 5 demonstrates the AFM images of St-P(CL-b-EO)-AgNPs and St-P(CL-b-EO)-AgNPs/Cp. The typical pale yellow colour of St-P(CL-b-EO)-AgNPs turns dark brown by the addition of Cp in it.

Various functionalities present on the Cp, viz. N-H, S-H, C-N, C=O, C-O have a tendency to donate lone pairs of electrons to AgNPs that makes the basis for strong interaction of the drug with the AgNPs. FTIR spectra of St-P(CL-b-EO), St-P(CL-b-EO)-AgNPs and St-P(CL-b-EO)-AgNPs/Cp demonstrate the possible mechanism of NP formation and their complexation with Cp, figure 6. A comparison of the FTIR spectra of St-P(CL-b-EO), and St-P(CL-b-EO)-AgNPs reveal that all the characteristic peaks for the St-P(CL-b-EO) are present after interaction with AgNPs; hence, the non-ionic St-P(CL-b-EO) sterically stabilized the AgNPs. Moreover, the characteristic peak at 1383 cm$^{-1}$ in the spectrum of

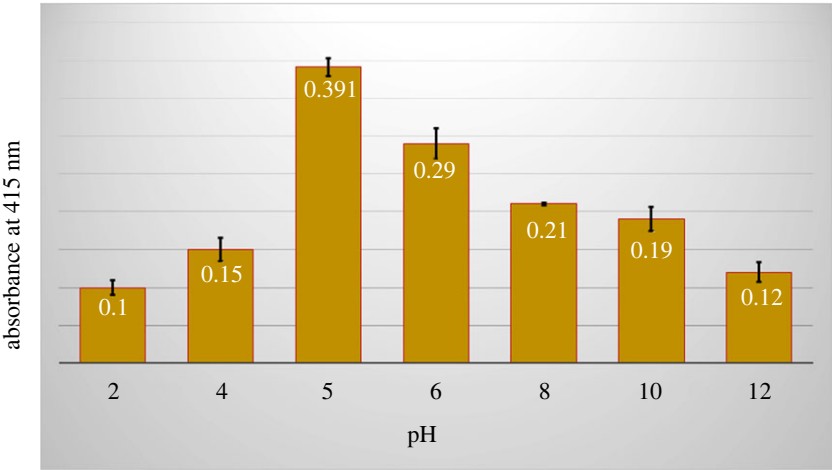

**Figure 4.** Change in the intensity of SPR band of St-P(CL-b-EO)-AgNPs as a function of pH.

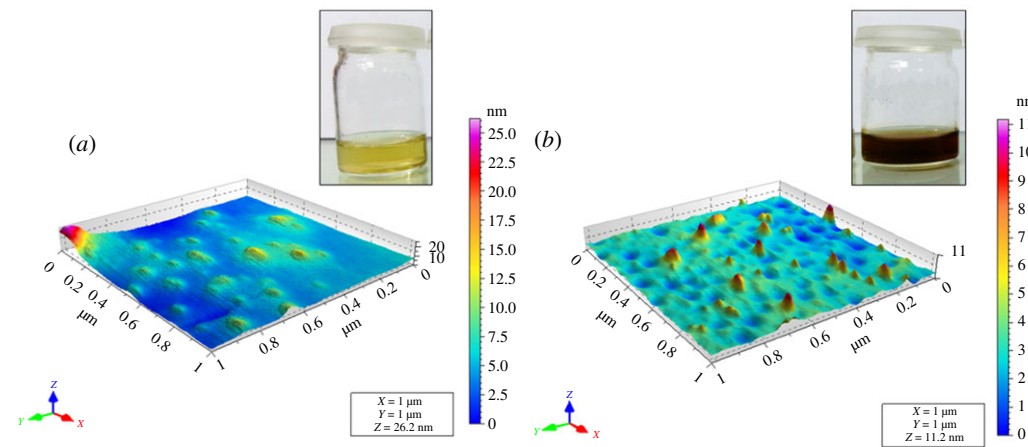

**Figure 5.** Atomic force micrographs (AFMs), (*a*) P(EO-b-CL)-AgNPs; (*b*) P(EO-b-CL)-AgNPs/Cp.

St-P(CL-b-EO)-AgNPs confirms the formation of AgNPs [42,43]. The FTIR spectrum of Cp has specific peaks at 3272, 2602, 1280 and 1162 cm$^{-1}$ for N-H stretch, S-H stretch, C-O stretch and C-N stretch, respectively. Interestingly, in the FTIR spectrum of St-P(CL-b-EO)-AgNPs/Cp, the S-H, C-N and N-H peaks of Cp have diminished while the peak at 1383 cm$^{-1}$ in St-P(CL-b-EO)-AgNPs (typical for AgNP) gets minimized. The shifting and diminishing of characteristic peaks of drug and the minimization of AgNPs peak indicate a strong affinity of Cp with AgNPs. Furthermore, size of the St-P(CL-b-EO)-AgNPs has considerably decreased while their stability increased after its complexation with Cp, as shown by zeta sizer and zeta potential studies [33].

## 3.4. Encapsulation efficiency of St-P(CL-b-EO)-silver nanoparticles for cephalexin

The amount of Cp encapsulated in St-P(CL-b-EO)-AgNPs was calculated by the difference between the total amount of Cp initially taken and free drug in the formulation after the complexation step, determined by UV spectrophotometry. The encapsulation efficiency of the formulation was found to be 65.26%.

## 3.5. Antibacterial analysis

### 3.5.1. Growth inhibition efficiency against *Staphylococcus aureus* (ATCC-25923)

The growth inhibition efficiency of Cp, St-P(CL-b-EO), St-P(CL-b-EO)-AgNPs and St-P(CL-b-EO)-AgNPs/Cp were appraised against *S. aureus* through tetrazolium microplate assay. All the test samples were incubated against the selected microorganism in a concentration range of 10–500 µg ml$^{-1}$ and their

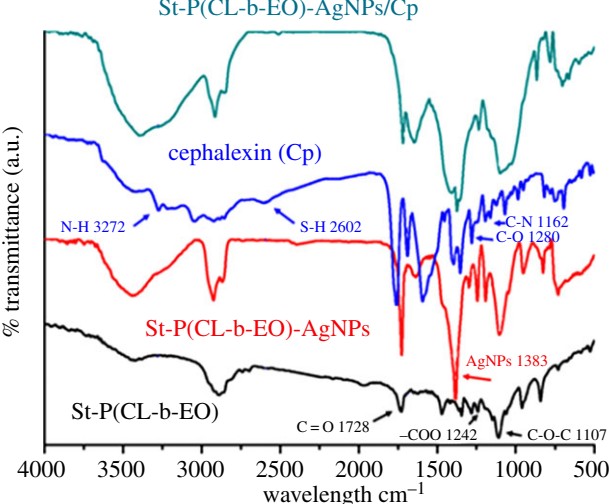

**Figure 6.** FTIR spectra of St-P(CL-b-EO), Cp, St-P(CL-b-EO)-AgNPs and St-P(CL-b-EO)-AgNPs/Cp complex [33].

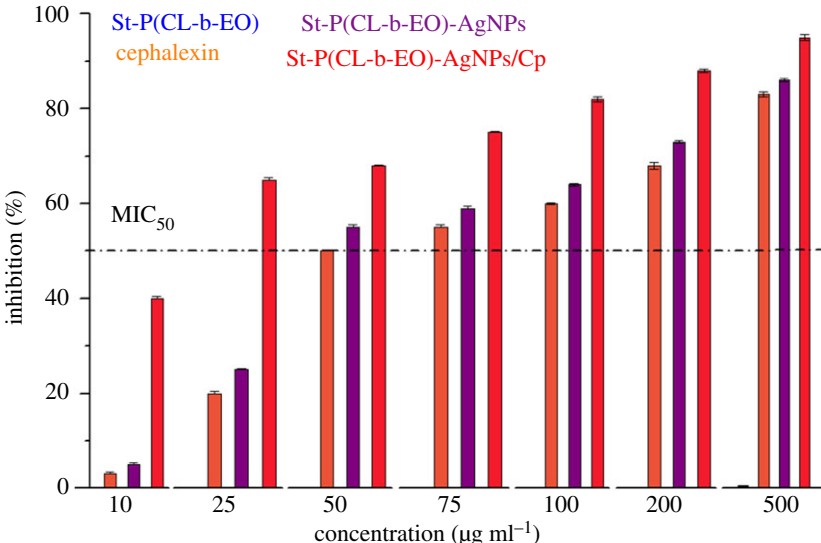

**Figure 7.** Growth inhibition efficiency as a function of concentration of St-P(CL-b-EO), cephalexin (Cp), St-P(CL-b-EO)-AgNPs and St-P(CL-b-EO)-AgNPs-Cp against *S. aureus*.

inhibition efficiency was evaluated. The results of inhibition efficiency as a function of concentration are summarized in figure 7. St-P(CL-b-EO) did not show any activity against *S. aureus* (ATCC-25923) with the highest tested concentration of 500 µg ml$^{-1}$. Independent administration of Cp and St-P(CL-b-EO)-AgNPs were able to achieve 50% inhibition efficacy (MIC$_{50}$) at 50 µg ml$^{-1}$ concentration. The inhibition efficiency was dramatically increased by delivery of Cp by its complexation with St-P(CL-b-EO)-AgNPs [St-P(CL-b-EO)-AgNPs/Cp]; the application of only 25 µg ml$^{-1}$ was able to inhibit the growth of *S. aureus* (ATCC-25923) to 65% (15% more than MIC$_{50}$). This enhanced growth inhibition efficiency is attributed to the synergistic combination of Cp with St-P(CL-b-EO)-AgNPs through complexation that enhances its transport through the bacterial cell wall owing to their adherence to proteins and their affinity for sulfur in cellular metabolism. The numerical data are presented in the electronic supplementary material, table S1.

### 3.5.2. Biofilm growth inhibition efficiency against *Staphylococcus aureus* (ATCC-25923)

Biofilms are microbial populations embedded in a self-producing matrix that develop on living and nonliving solid surfaces. Biofilm inhibition is considered as a major drug target for the treatment of

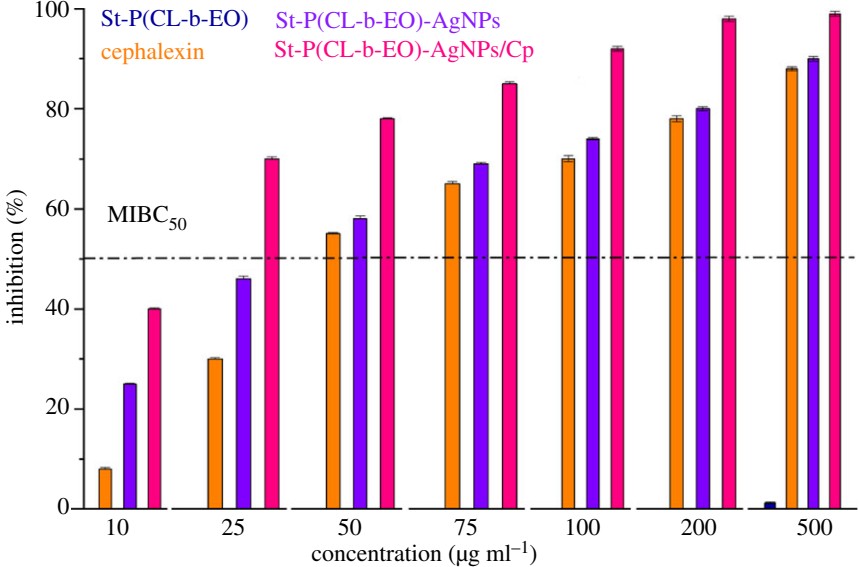

**Figure 8.** Biofilm growth inhibition efficiency as a function of concentration of St-P(CL-b-EO), cephalexin (Cp), St-P(CL-b-EO)-AgNPs and St-P(CL-b-EO)-AgNPs/Cp against *S. aureus*.

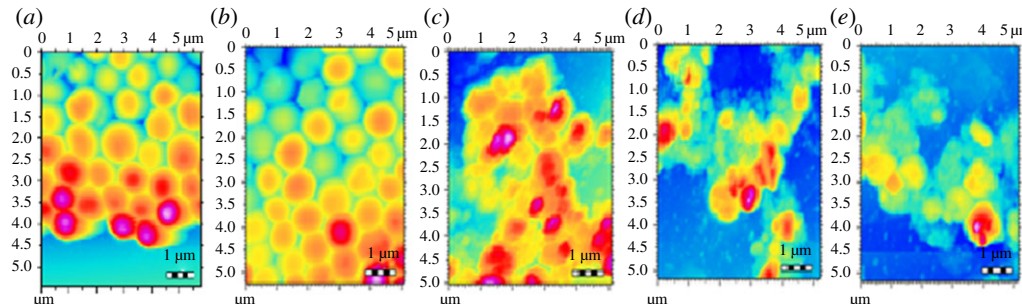

**Figure 9.** AFM topographic images of *S. aureus* bacterial cells: (*a*) control; after treatment at $MIC_{50}$ with (*b*) St-P(CL-b-EO), (*c*) Cp, (*d*) St-P(CL-b-EO)-AgNPs, (*e*) St-P(CL-b-EO)-AgNPs-Cp.

various microbial infections. An increased biofilm resistance to conventional treatments paves the way for the development of new strategies for efficient drug administration. The improvement in the antibacterial activity of Cp by delivery through St-P(CL-b-EO)-AgNPs/Cp is further endorsed by assay against biofilm growth inhibition efficiency, figure 8. St-P(CL-b-EO) did not show any antibiofilm activity against the target bacterial strain. Cp achieved 54% biofilm inhibition efficacy ($MIBC_{50}$) at a concentration of 50 μg ml$^{-1}$, whereas biofilm inhibition efficiency of 58% is achieved by St-P(CL-b-EO)-AgNPs at a concentration of 50 μg ml$^{-1}$ ($MIBC_{50}$). The biofilm inhibition efficiency of a complex of Cp with St-P(CL-b-EO)-AgNPs [St-P(CL-b-EO)-AgNPs/Cp] is enhanced several-fold, and more than 70% of inhibition was achieved while using only 25 μg ml$^{-1}$. Enhanced antibiofilm activity of St-P(CL-b-EO)-AgNPs loaded Cp is attributed to the penetrative nature of the drug cargo St-P(CL-b-EO)-AgNPs through the bacterial cell wall. The numerical data are presented in the electronic supplementary material, table S2.

### 3.5.3. Topographic imaging of *Staphylococcus aureus* colonies by atomic force microscopy

AFM was employed for visualization of the destruction of the bacterial cell walls and colonies of *S. aureus* by Cp, St-P(CL-b-EO), St-P(CL-b-EO)-AgNPs and St-P(CL-b-EO)-AgNPs/Cp at their $MIC_{50}$ and $MIBC_{50}$. The control cells of *S. aureus* have smooth, healthy and organized cell walls, figure 9*a*. St-P(CL-b-EO) did not cause any changes in the bacterial cell wall, thus confirming its inactivity towards the tested bacterial strain, figure 9*b*. The bacterial cells were denatured while treated with Cp at its $MIC_{50}$ (50 μg ml$^{-1}$), figure 9*c*. Similarly, St-P(CL-b-EO)-AgNPs at its $MIC_{50}$ (50 μg ml$^{-1}$) was able to achieve comparable

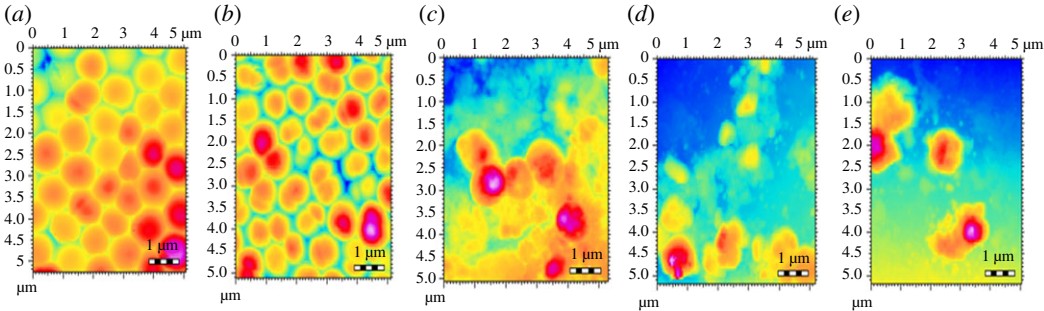

**Figure 10.** AFM topographic images of *S. aureus* bacterial cells biofilms: (*a*) control; after treatment at MIBC$_{50}$ with (*b*) P(EO-b-CL), (*c*) Cp, (*d*) P(EO-b-CL)-AgNPs, (*e*) P(EO-b-CL)-AgNPs-Cp.

destruction, figure 9*d*. The drug activity enhanced upon its loading in St-P(CL-b-EO)-AgNPs and complete destruction of the bacterial population is achieved at 25 µg ml$^{-1}$ (MIC$_{50}$). The bacterial population was converted into molten mass-like structures, figure 9*e*. The AFM analysis of biofilm inhibition efficiency has also endorsed the assay results as shown by topographic imaging of the biofilms after treatment with the drug and its conjugates at their respective MIBC$_{50}$, figure 10.

## 4. Conclusion

In this study, St-P(CL-b-EO)-AgNPs were employed as an enhancer in the antibacterial activity of Cp through complexation. The synergistic effect of complexation of St-P(CL-b-EO)-AgNPs with Cp resulted in reduction in the MIC$_{50}$ and MIBC$_{50}$ by at least one order of magnitude. The visual destruction of bacterial cell walls and their biofilms was clearly observed by application of the drug and its conjugates at their respective MIC$_{50}$ and MIBC$_{50}$ through AFM topographic imaging. Hence, the co-formulation of antibiotics with NPs provides an excellent tool to counter the unresolved problem of increasing resistance of pathogenic bacteria against common antibiotics. Furthermore, the combination of antibiotic with NPs could be a realistic approach for the reduction in the amount of antibiotics use. The study indicates the high efficiency of drug encapsulation by polymer-stabilized AgNPs. Furthermore, the inherent possibility of polymers for tailoring their total molar mass, molar mass of individual segments, chemical composition and architecture, opens new horizons for further enhancement in drug efficacy.

Data accessibility. Electronic supplementary material contains SEC elugrams, and NMR spectra of precursors and final product. The numerical data of the growth inhibition efficiency and biofilm growth film inhibition is also included.
Authors' contributions. S.R. did NP studies and wrote the manuscript; S.P. did antibacterial assay studies and helped write that part; S.A. supervised the antibacterial studies part; M.R.S. supervised NP formation and the characterization part; M.I.M. conceived the idea and finalized the manuscript.
Competing interests. The authors declare no conflict of interest.
Funding. This research is supported by the Higher Education Commission of Pakistan under 'National Research Program for Universities' against project no. 5754.

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
