## [Reviewer comments · Royal Society Open Science]

Review History

RSOS-201097.R0 (Original submission)

Review form: Reviewer 1

Is the manuscript scientifically sound in its present form?

Yes

Are the interpretations and conclusions justified by the results?

Yes

Is the language acceptable?

Yes

Do you have any ethical concerns with this paper?

No

Have you any concerns about statistical analyses in this paper?

Yes

Recommendation?

Accept with minor revision (please list in comments)

Comments to the Author(s)

The antibacterial activity of the silver nanoparticles (AgNPs) can be stabilized with 4-arm star-shaped poly(ethylene oxide)-block-poly(ϵ -caprolactone) copolymer [St-P(CL-b-EO)]. The manuscript is well organized, so I would like to ask the authors to revise the following minor points. This work collects excellent data on the target subjects.

How about the materials' stability? This point is very critical if we consider practical applications.

The authors show wide-angle XRD patterns for samples. How about the average crystallite sizes? This size is matched with TEM data?

Related papers have been reported by different research groups. It is better to cite the following refs to support some related paragraphs in the introduction part.

Nanoporous/mesoporous metals have been reported by soft- and hard-templating methods (JACS, 140, 12434, 2018; Chem. Sci., 2019, 10, 4054, etc.). Other different groups have reported several reviews which are useful for readers (Nano Today 21, 91 (2018); Acc. Chem. Res. 51, 1764 (2018), etc).

Overall the manuscript is well written, but I want to see the authors' perspective (future vision) on this research in the conclusion part.

Review form: Reviewer 2

Is the manuscript scientifically sound in its present form?

Yes

Are the interpretations and conclusions justified by the results?

Yes

Is the language acceptable?

Yes

Do you have any ethical concerns with this paper?

No

Have you any concerns about statistical analyses in this paper?

No

Recommendation?

Major revision is needed (please make suggestions in comments)

Comments to the Author(s)

Silver nanoparticles (AgNPs) were prepared by stabilization with 4-arm star-shaped block copolymer [St-P(CL-b-EO)] and its application as drug delivery vehicle for cephalexin (Cp) was evaluated against pathogenic Staphylococcus aureus. The manuscript can be accepted for publication after issuing the following considerations.

(1) The characterization data of St-P(CL-b-EO) sample used in this work should be provided in the supporting information.

- (2) The authors should discuss why they choose the block ratio with star-PCL10K and MeO-PEO2K?
- (3) The morphology of nanoparticles should also be characterized by TEM.
- (4) The loading efficiency of drug should be determined.
- (5) The stability of material should also be studied by DLS test to monitor the size change.

Decision letter (RSOS-201097.R0)

Dear Dr Malik:

Title: Enhancement in the Antibacterial Activity of Cephalexin by its Delivery through Star-shaped Poly(ϵ -caprolactone)-block-poly(ethylene oxide)-AgNPs
Manuscript ID: RSOS-201097

The editor assigned to your manuscript has now received comments from reviewers. We would like you to revise your paper in accordance with the referee and Subject Editor suggestions which can be found below (not including confidential reports to the Editor). Please note this decision does not guarantee eventual acceptance.

Please submit your revised paper before 02-Sep-2020. Please note that the revision deadline will expire at 00.00am on this date. If we do not hear from you within this time then it will be assumed that the paper has been withdrawn. In exceptional circumstances, extensions may be possible if agreed with the Editorial Office in advance. We do not allow multiple rounds of revision so we urge you to make every effort to fully address all of the comments at this stage. If deemed necessary by the Editors, your manuscript will be sent back to one or more of the original reviewers for assessment. If the original reviewers are not available we may invite new reviewers.

On behalf of the Subject Editor Professor Anthony Stace and the Associate Editor Dr Chaohua Cui.

RSC Associate Editor:
Comments to the Author:
(There are no comments.)

RSC Subject Editor:
Comments to the Author:
(There are no comments.)

Reviewers' Comments to Author:
Reviewer: 1

Comments to the Author(s)
The antibacterial activity of the silver nanoparticles (AgNPs) can be stabilized with 4-arm star-shaped poly(ethylene oxide)-block-poly(ϵ -caprolactone) copolymer [St-P(CL-b-EO)]. The manuscript is well organized, so I would like to ask the authors to revise the following minor points. This work collects excellent data on the target subjects.

How about the materials' stability? This point is very critical if we consider practical applications.

The authors show wide-angle XRD patterns for samples. How about the average crystallite sizes? This size is matched with TEM data?

Related papers have been reported by different research groups. It is better to cite the following refs to support some related paragraphs in the introduction part.

Nanoporous/mesoporous metals have been reported by soft- and hard-templating methods (JACS, 140, 12434, 2018; Chem. Sci., 2019, 10, 4054, etc.). Other different groups have reported several reviews which are useful for readers (Nano Today 21, 91 (2018); Acc. Chem. Res. 51, 1764 (2018), etc).

Overall the manuscript is well written, but I want to see the authors' perspective (future vision) on this research in the conclusion part.

Reviewer: 2

Comments to the Author(s)
Silver nanoparticles (AgNPs) were prepared by stabilization with 4-arm star-shaped block copolymer [St-P(CL-b-EO)] and its application as drug delivery vehicle for cephalexin (Cp) was evaluated against pathogenic Staphylococcus aureus. The manuscript can be accepted for publication after issuing the following considerations.

- (1) The characterization data of St-P(CL-b-EO) sample used in this work should be provided in the supporting information.
- (2) The authors should discuss why they choose the block ratio with star-PCL10K and MeO-PEO2K?
- (3) The morphology of nanoparticles should also be characterized by TEM.
- (4) The loading efficiency of drug should be determined.
- (5) The stability of material should also be studied by DLS test to monitor the size change.

Author's Response to Decision Letter for (RSOS-201097.R0)

See Appendix A.

RSOS-201097.R1 (Revision)

Review form: Reviewer 1

Is the manuscript scientifically sound in its present form?

Yes

Are the interpretations and conclusions justified by the results?

Yes

Is the language acceptable?

Yes

Do you have any ethical concerns with this paper?

No

Have you any concerns about statistical analyses in this paper?

Yes

Recommendation?

Accept as is

Comments to the Author(s)

Improved.

Review form: Reviewer 2

Is the manuscript scientifically sound in its present form?

Yes

Are the interpretations and conclusions justified by the results?

Yes

Is the language acceptable?

Yes

Do you have any ethical concerns with this paper?

No

Have you any concerns about statistical analyses in this paper?

No

Recommendation?

Accept with minor revision (please list in comments)

Comments to the Author(s)

The authors have successfully addressed most of the concerns raised by the reviewers, thereby I think it can be considered for publication. Before that, I still hold the opinion that the authors should put all the DETAILED characterization data of polymers in the ESI, including SEC, ¹H NMR, etc.

Decision letter (RSOS-201097.R1)

Dear Dr Malik:

Title: Enhancement in the Antibacterial Activity of Cephalexin by its Delivery through Star-shaped Poly(ϵ -caprolactone)-block-poly(ethylene oxide)-AgNPs
Manuscript ID: RSOS-201097.R1

Thank you for submitting the above manuscript to Royal Society Open Science. On behalf of the Editors and the Royal Society of Chemistry, I am pleased to inform you that your manuscript will be accepted for publication in Royal Society Open Science subject to minor revision in accordance with the referee suggestions. Please find the reviewers' comments at the end of this email.

The reviewers and handling editors have recommended publication, but also suggest some minor revisions to your manuscript. Therefore, I invite you to respond to the comments and revise your manuscript.

Because the schedule for publication is very tight, it is a condition of publication that you submit the revised version of your manuscript before 11-Sep-2020. Please note that the revision deadline will expire at 00.00am on this date. If you do not think you will be able to meet this date please let me know immediately.

Kind regards,
Dr Ellis Wilde
Publishing Editor, Journals

On behalf of the Subject Editor Professor Anthony Stace and the Associate Editor Dr Chaohua Cui.

RSC Subject Editor
Comments to the Author:
(There are no comments.)

RSC Associate Editor
Comments to the Author:
(There are no comments.)

Reviewer comments to Author:

Reviewer: 1

Comments to the Author(s)

Improved.

Reviewer: 2

Comments to the Author(s)

The authors have successfully addressed most of the concerns raised by the reviewers, thereby I think it can be considered for publication. Before that, I still hold the opinion that the authors should put all the DETAILED characterization data of polymers in the ESI, including SEC, ¹H NMR, etc.

Author's Response to Decision Letter for (RSOS-201097.R1)

See Appendix B.

Decision letter (RSOS-201097.R2)

Dear Dr Malik:

Title: Enhancement in the Antibacterial Activity of Cephalexin by its Delivery through Star-shaped Poly(ϵ -caprolactone)-block-poly(ethylene oxide)-AgNPs

Manuscript ID: RSOS-201097.R2

It is a pleasure to accept your manuscript in its current form for publication in Royal Society Open Science. The chemistry content of Royal Society Open Science is published in collaboration with the Royal Society of Chemistry.

On behalf of the Subject Editor Professor Anthony Stace and the Associate Editor Dr Chaohua Cui.

RSC Associate Editor
Comments to the Author:
(There are no comments.)

Reviewer(s)' Comments to Author:

Appendix A

Response to the Reviewer's Comments (RSOS-201097)

Reviewer 1:

How about the materials' stability? This point is very critical if we consider practical applications.

The stability of the St-P(CL-b-EO)-AgNPs before and after complexation with drug is discussed in detail in the manuscript. Relevant text from manuscript is

The stability of St-P(CL-b-EO)-AgNPs against different external parameters such as temperature, presence of electrolytes and pH are evaluated. Temperature treatment of synthesized St-P(CL-b-EO)-AgNPs at 100 °C for 20 minutes resulted in an increase in the intensity of SPR band which is an indication of enhanced stabilization (**Error! Reference source not found.**). Moreover, St-P(CL-b-EO)-AgNPs persisted for more than 12 months at ambient temperature.

To check the stability of St-P(CL-b-EO)-AgNPs in presence of electrolytes, various electrolyte concentrations (0.01 M – 5 M NaCl) were added in St-P(CL-b-EO)-AgNPs and their stability is monitored by UV-visible spectroscopy, **Error! Reference source not found.** A visible decrease in the sharpness of the absorbance peak is observed by increase in the electrolyte concentration. Moreover, the typical AgNPs peak just appeared as a shoulder by addition of NaCl beyond 1.0 M. The aggregation of AgNPs by addition of electrolyte is attributed to presence of large amount of Cl^{-1} ions in the solution.^{35, 36}

Synthesized St-P(CL-b-EO)-AgNPs were slightly acidic in nature having pH~5. The changes in the pH in a range of 2–12 resulted in a visible reduction in the intensity of the SPR band of St-P(CL-b-EO)-AgNPs, **Error! Reference source not found.** It is found that synthesized St-P(CL-b-EO)-AgNPs have maximum stability at original pH (~5), as deduced from the intensity of SPR band. Furthermore, stability of AgNPs decreased while moving away from original pH, which is more pronounced towards acidic environment compared to neutral or slightly basic pH.

The addition of Cp in the St-P(CL-b-EO)-AgNPs resulted in formation of a complex that is more stable compared to parent St-P(CL-b-EO)-AgNPs. The size of St-P(CL-b-EO)-AgNPs/Cp complex decreased (DLS & AFM analysis) while stability increased (zeta potential values) compared to St-P(CL-b-EO)-AgNPs.[30] The average size of St-P(CL-b-EO)-AgNPs decreased from 151.5 ± 8.071 to 140.5 ± 9.895 nm for St-P(CL-b-EO)-AgNPs/Cp. Similarly, zeta potential of values of -3.33 ± 4.18 for St-P(CL-b-EO)-AgNPs compared to -10.6 ± 4.27 mV for St-P(CL-b-EO)-AgNPs/Cp indicates enhanced stability of latter.

The authors show wide-angle XRD patterns for samples. How about the average crystallite sizes? This size is matched with TEM data?

I think reviewer is mistaken, we have not presented any XRD patterns in the manuscript. Instead, AFM images are presented. As complementary techniques, we have used DLS (zetasizer) and zeta potential. The outcome of all the techniques endorse each other.

Related papers have been reported by different research groups. It is better to cite the following refs to support some related paragraphs in the introduction part.

Nanoporous/mesoporous metals have been reported by soft- and hard-templating methods (JACS, 140, 12434, 2018; Chem. Sci., 2019, 10, 4054, etc.). Other different groups have reported several reviews which are useful for readers (Nano Today 21, 91 (2018); Acc. Chem. Res.51,1764(2018),etc).

The recommended papers along with couple of other reviews are now cited in the manuscript as per suggestion of the reviewer

Overall the manuscript is well written, but I want to see the authors' perspective (future vision) on this research in the conclusion part.

Following phrase is added in the conclusion

Furthermore, the inherent possibility of polymers for tailoring their total molar mass, molar mass

of individual segments, chemical composition, and architecture, opens new horizons for further enhancement in the drug efficacy.

Reviewer: 2

(1) The characterization data of St-P(CL-b-EO) sample used in this work should be provided in the supporting information.

The characterization data is now added in the main document along with reference for further details as

Molecular Characterization of St-P(CL-b-EO)

As mentioned in the experimental section, St-P(CL-b-EO) was synthesized by coupling of St-PCL_{10K} with MeO-PEO_{2K}. The number average and weight average molar mass of synthesized 4 arm St-PCL_{10K} as obtained by polystyrene calibrated size exclusion chromatography was 9700 and 13700 g/mol. MeO-PEO_{2K} used in coupling reaction was commercial product. The 4 arm St-P(CL-b-EO) were comprehensively characterized by size exclusion chromatography, FTIR, ¹H-NMR, liquid chromatography at critical conditions of both PCL and PEO. The comprehensive characterization of intermediates and final product clearly demonstrate the successful synthesis which is elaborated in great detail in ref. ³⁸

(2) The authors should discuss why they choose the block ratio with star-PCL10K and MeO-PEO2K?

We selected the block ratio of star-PCL10K and MeO-PEO2K just to keep the ratio of same content of hydrophobic and hydrophilic segments. In further studies, we plan to compare the selectivity of the block lengths, their ratios of star block copolymer in context of their efficiency as antibacterial material.

(3) The morphology of nanoparticles should also be characterized by TEM.

In this study morphology and size of the NPs is studied through AFM. As a complementary technique we employed zetasizer (DLS) and zeta potential to confirm the size, size distribution, and stability of NPs before and after encapsulation with drug. The relevant discussion in the manuscript is given below

The addition of Cp in the St-P(CL-b-EO)-AgNPs resulted in formation of a complex that is more stable compared to parent St-P(CL-b-EO)-AgNPs. The size of St-P(CL-b-EO)-AgNPs/Cp complex decreased (DLS & AFM analysis) while stability increased (zeta potential values) compared to St-P(CL-b-EO)-AgNPs.²⁷ The average size of St-P(CL-b-EO)-AgNPs decreased from 151.5 ± 8.071 to 140.5 ± 9.895 nm for St-P(CL-b-EO)-AgNPs/Cp. Similarly, zeta potential of values of -3.33 ± 4.18 for St-P(CL-b-EO)-AgNPs compared to -10.6 ± 4.27 mV for St-P(CL-b-EO)-AgNPs/Cp indicates enhanced stability of latter. **Error! Reference source not found.** demonstrates the AFM images of St-P(CL-b-EO)-AgNPs and St-P(CL-b-EO)-AgNPs/Cp. Typical pale yellow color of St-P(CL-b-EO)-AgNPs turns dark brown by addition of cephalexin in it.

(4) The loading efficiency of drug should be determined.

Loading efficiency of the synthesized NPs is now added in the manuscript. The additions in the manuscript in this connection are

. Experimental Section:

Determination of Loading Efficiency

Loading efficiency of drug in the above-mentioned complexation is determined through UV visible spectrophotometer (Shimadzu, UV-240, Hitachi U-3200). In brief, above-mentioned formulation of drug and nanoparticles was centrifuged at 14000 rpm for 25 min. The pellets at the bottom of the Eppendorf tube were carefully collected, followed by dissolution of these pellets in acetone. In parallel, a calibration curve for the amount of cephalexin in acetone is constructed by recording UV visible spectra at 266 nm of solutions of different concentrations of drug (0.1 to 0.5 mg/mL) which is then employed for determination of free drug in acetone. The encapsulation efficiency of micelles was calculated by using following equation.

$$\%Encapsulation\ Efficiency = \frac{A_{(Total\ drug)} - A_{(Free\ drug)}}{A_{(Total\ drug)}} \times 100$$

Where A is the amount of the drug

Result and discussion:

Cephalexin molecules were gradually entrapped into the hydrophobic core of St-P(CL-b-EO)-nanoparticles via self-assembly. The amount of cephalexin encapsulated into polymeric nanoparticles was calculated by difference between total amount of cephalexin used for encapsulation in nanoparticles and amount of free cephalexin in the formulation, determined by UV spectrophotometry after removing free cephalexin and cephalexin bounded on the surface of polymeric nanoparticles by centrifugation. The encapsulation efficiency was found to be 65.26%. results indicate that we can achieve higher drug loading by using star shaped polymer.

(5) The stability of material should also be studied by DLS test to monitor the size change.

This is already discussed in the manuscript as

The addition of Cp in the St-P(CL-b-EO)-AgNPs resulted in formation of a complex that is more stable compared to parent St-P(CL-b-EO)-AgNPs. The size of St-P(CL-b-EO)-AgNPs/Cp complex decreased (DLS & AFM analysis) while stability increased (zeta potential values) compared to St-P(CL-b-EO)-AgNPs.²⁷ The average size of St-P(CL-b-EO)-AgNPs decreased from 151.5 ± 8.071 to 140.5 ± 9.895 nm for St-P(CL-b-EO)-AgNPs/Cp. Similarly, zeta potential of values of -3.33 ± 4.18 for St-P(CL-b-EO)-AgNPs compared to -10.6 ± 4.27 mV for St-P(CL-b-EO)-AgNPs/Cp indicates enhanced stability of latter. **Error! Reference source not found.** demonstrates the AFM images of St-P(CL-b-EO)-AgNPs and St-P(CL-b-EO)-AgNPs/Cp. Typical pale yellow color of St-P(CL-b-EO)-AgNPs turns dark brown by addition of cephalexin in it.

Appendix B

Response to the Reviewer's Comments (RSOS-201097R1)

Reviewer: 2

Comments to the Author(s)

The authors have successfully addressed most of the concerns raised by the reviewers, thereby I think it can be considered for publication. Before that, I still hold the opinion that the authors should put all th DETAILED characterization data of polymers in the ESI, including SEC, ¹H NMR, etc.

SEC elugrams and ¹H-NMR spectra of precursors MeO-PEO_{2K}, and St-PCL_{10K}, along with the final product St-P(CL_{10K}-b-EO_{2K}) are presented in supplementary material.